

# Nonlinear response in diffusive systems

Luca V. Delacrétaz[1,2,3] and Ruchira Mishra[1]

**1** Kadanoff Center for Theoretical Physics, University of Chicago, Chicago, IL 60637, USA
**2** James Franck Institute, University of Chicago, Chicago, IL 60637, USA
**3** Department of Theoretical Physics, Université de Genève, 1211 Genève, Switzerland

## Abstract

Nonintegrable systems thermalize, leading to the emergence of fluctuating hydrodynamics. Typically, this hydrodynamics is diffusive. We use the effective field theory (EFT) of diffusion to compute higher-point functions of conserved densities. We uncover a simple scaling behavior of correlators at late times, and, focusing on three and four-point functions, derive the asymptotically exact universal scaling functions that characterize nonlinear response in diffusive systems. This allows for precision tests of thermalization beyond linear response in quantum and classical many-body systems. We confirm our predictions in a classical lattice gas.

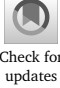

# 1 Introduction

Most interacting systems thermalize at nonzero temperature. Their late time dynamics is then described by the fluctuations of conserved densities, or hydrodynamics, and their global equilibration towards thermal equilibrium.

In the case where a single density is conserved, the hydrodynamic description is typically diffusive [1], with density two-point function taking a universal form

$$\langle n(x,t)n(0,0)\rangle = \frac{\chi T}{(4\pi D|t|)^{d/2}}e^{-x^2/(4D|t|)} + O\left(1/t^{\frac{d}{2}+1}\right) + O(1/t^d),\qquad(1)$$

which is entirely fixed at late times up to two non-universal constants: the charge susceptibility $\chi \equiv \frac{dn}{d\mu}$, and the diffusivity $D$. Here $T$ is the temperature and $d$ the spatial dimension. The subleading corrections denote higher derivative [2] and loop contributions [3] respectively – both are irrelevant and lead to small corrections at late times. This two-point function captures linear response in all diffusive systems and organizes it in terms of a few non-universal 'Wilsonian coefficients' of the effective field theory (EFT) of diffusion [4].

Diffusive systems feature nonlinear response as well. For example, a diffusivity $D = D(n)$ or a susceptibility $\chi = \chi(n)$ that is a nontrivial function of the background density immediately implies nonlinear response. More generally, the EFT of diffusion has interactions that do not have simple interpretations as nonlinearities of transport or thermodynamic parameters (see, e.g., [5]); we will show however that to leading order at late times, nonlinear response is simply controlled by the coefficients in the expansion of $D(n)$ and $\sigma(n) \equiv \chi(n)D(n)$ around the background value of the density

$$D(n) = D + D'\delta n + \frac{1}{2}D''\delta n^2 + \cdots, \qquad \sigma(n) = \sigma + \sigma'\delta n + \frac{1}{2}\sigma''\delta n^2 + \cdots.\qquad(2)$$

The EFT predictions for nonlinear response therefore do not involve, to leading order, any Wilsonian coefficients beyond those measurable within linear response at several values of the background density.

In this paper, we present a simple scaling argument for the late time behavior of equilibrium connected $N$-point functions in diffusive systems $\langle n((N-1)t)\cdots n(2t)n(t)n(0)\rangle \sim 1/t^{(N-1)d/2}$. Using the EFT of diffusion [4], we confirm this scaling and, focusing on $N = 3$, obtain the entire three-point function $\langle n(x_2,t_2)n(x_1,t_1)n(0,0)\rangle$ at late times, which is the sum of two universal scaling functions multiplied by the nonuniversal parameters $D'$ and $\sigma'$ from (2). The EFT results are asymptotically exact: the only approximation lies in the late time expansion of correlators. Finally, we numerically study a classical lattice gas model and find excellent agreement with our predictions.

One motivation for this work are recent experimental developments in probing the nonlinear response of correlated many-body systems [6–8], and particularly progress in studying higher-harmonic generation from low-frequency (single THz) sources [9,10], which may be governed by hydrodynamics when $\hbar\omega \lesssim T$. Our results should apply to diffusive metals, including anomalous metals [11] and strange metals [12,13], as well as to diffusion in correlated insulators [14]. We further hope that our results can serve as benchmarks for precision tests of thermalization (or lack thereof) in numerics.

Nonlinear response in thermal states has been studied for some time. Generalized fluctuation-dissipation or KMS relations were found in [15,16]. Certain general properties were also studied recently through the lens of the eigenstate thermalization hypothesis in [17,18]. Higher order correlation functions were computed in holographic models [19,20], models with relaxational dynamics [19,21], integrable systems [22,23] as well as more general ballistic regimes [24]. Nonlinear response is also related to higher cumulants (or full counting statistics) in out-of-equilibrium states (see, e.g., [25–27]).

## 2 General scaling considerations

A simple scaling argument gives the general form of nonlinear response in equilibrium: From (1), one finds that densities scale as $\delta n \sim 1/x^{d/2}$ and $x^2 \sim Dt$. Nonlinear response however requires corrections to diffusive scaling from irrelevant interactions. These leading non-Gaussianities are suppressed by fluctuations $\delta n \sim 1/x^{d/2}$: a three-point function for example will therefore scale as $\langle nnn \rangle \sim x^{-3d/2} x^{-d/2} \sim x^{-2d}$. A connected $N$-point function will require $N-2$ such non-Gaussianities and therefore scale as $\langle n \cdots n \rangle \sim x^{-Nd/2} x^{-(N-2)d/2}$, or

$$\langle n(x_{N-1}, t_{N-1}) \cdots n(x_1, t_1) n(0,0) \rangle = \frac{\chi T}{(D\bar{t})^{(N-1)d/2}} g_N\left(\frac{t_i}{t_j}, y_i\right) + \cdots, \quad y_i \equiv \frac{x_i}{\sqrt{Dt_i}}, \quad (3)$$

where $g_N$ is a universal dimensionless scaling function (up to a finite number of Wilsonian coefficients) depending on cross-ratios of coordinates. The factor of $\chi T$ can be obtained from dimensional analysis. We have chosen to parametrize the overall scaling dependence in terms of the geometric mean of the differences in times (assuming that $t_{N-1} > \cdots t_2 > t_1 > 0$):

$$\bar{t} = [(t_{N-1} - t_{N-2}) \cdots (t_2 - t_1)(t_1 - 0)]^{1/(N-1)}. \quad (4)$$

The subleading corrections in (3) are similar to those in (1): higher derivative contributions give corrections with a relative $O(1/\bar{t})$ suppression, and loops give corrections with a relative $O(1/\bar{t}^{d/2})$ suppression (up to logarithms).

Eq. (1) shows that the two-point function ($N = 2$) indeed takes the form (3), with

$$g_2\left(\frac{x^2}{Dt}\right) = \frac{1}{(4\pi)^{d/2}} e^{-\frac{1}{4}\frac{x^2}{Dt}}, \quad (5)$$

the universal scaling function describing linear response in diffusive systems. We study higher-point functions $N \geq 3$ below, focusing particularly on $N = 3, 4$. We will obtain the universal form of the density three-point function in diffusive systems

$$\langle n(x_2, t_2) n(x_1, t_1) n(0,0) \rangle = \frac{\chi T}{[D\sqrt{t_1(t_2 - t_1)}]^d} g_3 + \cdots. \quad (6)$$

We will find that the cubic scaling function $g_3$ separates into two universal scaling functions, with non-universal coefficients proportional to $D' \equiv dD(n)/dn$ and $\sigma' \equiv d\sigma(n)/dn$:

$$g_3 = \chi T \frac{D'}{D} g_{3,D'}\left(\frac{t_2}{t_1}, y_1, y_2\right) + \chi T \frac{\sigma'}{\sigma} g_{3,\sigma'}\left(\frac{t_2}{t_1}, y_1, y_2\right), \quad (7)$$

with $y_i \equiv x_i/\sqrt{Dt_i}$. Since both $D'$ and $\sigma'$ can be independently extracted from linear response measurements at various values of the density $n$, the EFT produces a prediction for the three-point function with no fitting parameters.

## 3 EFT calculation of higher-point functions

To compute the nonlinear response of diffusive systems, we use the EFT of diffusion developed by Crossley, Glorioso and Liu [4], see [28] for a review and [29, 30] for related work. It differs from previous approaches to fluctuating hydrodynamics [31, 32] and macroscopic fluctuation theory [26] in that it provides a systematic controlled expansion in fluctuations. In particular, it captures general nonlinearities in the noise field that are not visible at the level of constitutive relations, and are missed in other approaches. However, these terms correspond to fairly

irrelevant operators in the EFT and only give subleading corrections to correlation functions.[1] We therefore expect that previous approaches could also be used to obtain nonlinear correlators $\langle n(t_{N-1}, x_{N-1}) \cdots n(t_1, x_1) n(0,0) \rangle$ to leading order at late times (although this has, to our knowledge, not been done). Nevertheless, in the spirit of using a *controlled* approach – namely one where corrections to the leading answer can be systematically obtained, order by order – we use the EFT of diffusion in this paper.

Consider the partition function of a microscopic system[2] with a conservation law $\partial_\mu j^\mu = 0$ ($\mu = 0, 1, \ldots, d$ runs over spacetime indices)

$$Z[A_\mu^1, A_\mu^2] \equiv \mathrm{Tr}\left(U[A^1] \rho_\beta U^\dagger[A^2]\right), \tag{8}$$

where $\rho_\beta = e^{-\beta H} / \mathrm{Tr}(e^{-\beta H})$ is the thermal density matrix, and

$$U[A] = \mathcal{T} \exp\left\{-i \int_{-\infty}^\infty dt \left(H - \int d^d x\, j^\mu A_\mu(t, x)\right)\right\}, \tag{9}$$

is the time-evolution operator of the system with density and current coupled to a source $A_\mu$. We have introduced two independent background sources $A_\mu^1, A_\mu^2$ in (8), one on each leg of the Schwinger-Keldysh contour, to generate correlation functions of the density $n \equiv j^0$ (or current density $j^i$) with various operator orderings.[3] For example, defining

$$A_{r\mu} \equiv \frac{1}{2}\left(A_\mu^1 + A_\mu^2\right), \qquad A_{a\mu} \equiv A_\mu^1 - A_\mu^2, \tag{10}$$

the symmetric Green's function of density is obtained from

$$\frac{1}{2} \mathrm{Tr}\left(\rho_\beta \{n(t, x), n(0,0)\}\right) = \frac{1}{i^2} \frac{\delta^2 \log Z}{\delta A_{a0}(t, x) \delta A_{a0}(0,0)} \equiv G_{rr}(t, x). \tag{11}$$

One can similarly define higher-point functions, for example:

$$G_{rrr}(t_2, x_2; t_1, x_1) \equiv \frac{1}{i^3} \frac{\delta^3 \log Z}{\delta A_{a0}(t_2, x_2) \delta A_{a0}(t_1, x_1) \delta A_{a0}(0,0)}, \tag{12}$$

with similar expressions for $G_{rra}$ and $G_{raa}$, see Eq. (A.8). These correspond to various orderings of three point functions of densities, see, e.g., [16].

The EFT of diffusion [4] consists in representing the partition function (8) by an effective Lagrangian $\mathcal{L}$ of hydrodynamic degrees of freedom

$$Z[A_\mu^1, A_\mu^2] \simeq \int Dn D\phi_a\, e^{i \int dt d^d x\, \mathcal{L}}, \tag{13}$$

where $n$ is the fluctuating density and $\phi_a$ is related to the corresponding noise field. The assumption of hydrodynamics is that the only long-lived excitations in thermalizing systems are conserved densities – the entire nonlocal aspect of $Z[A^1, A^2]$ can therefore be realized through a local effective action of these long-lived degrees of freedom. Appendix A reviews the construction of this effective action. To leading order in the diffusive scaling, the nonlinear action takes the form

$$\mathcal{L} = \sigma(n) B_{ai}(i T B_{ai} - E_{r,i}) + B_{a0} n - D(n) B_{ai} \partial_i n + \cdots, \tag{14}$$

---

[1]The leading such terms give relative $1/t^{d+1}$, $1/t^{d/2+1}$, and $1/t$ corrections to the correlators of $N = 2, 3$ and $\geq 4$ densities respectively [5].

[2]While we use notation appropriate for quantum systems, our approach describes nonlinear response of thermalizing classical systems as well.

[3]Correlation functions of other microscopic operators can also be obtained from the EFT up to further Wilsonian coefficients, through operator matching equations [33]. See [34] for examples in classical spin chains.

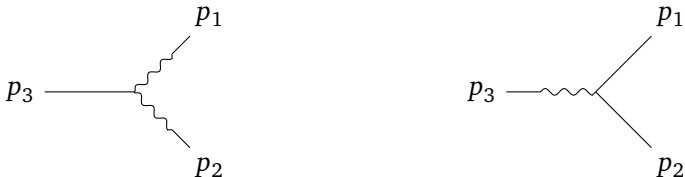

Figure 1: Two diagrams give contributions, proportional to $\sigma'$ and $D'$ respectively, to the density three-point function $G_{rrr}(p_2, p_1)$.

where $T = 1/\beta$ is the temperature, $B_{a\mu} = \partial_\mu \phi_a + A_{a\mu}$, and $E_{r,i} = \partial_0 A_{ri} - \partial_i A_{r0}$. The ellipses contain higher derivative terms, which give $O(q^2)$ corrections to observables, as well as further nonlinear terms that are more irrelevant than the leading nonlinear terms. To leading order, all nonlinearities come from expanding $\sigma(n)$ and $D(n)$ around the background value of the density as in (2). For example, up to cubic order in fields, turning off background fields $A_\mu$ momentarily, one has

$$\mathcal{L} = iT\sigma(\nabla\phi_a)^2 - \phi_a(\dot{n} - D\nabla^2 n) + iT\sigma' n(\nabla\phi_a)^2 + \frac{1}{2}D' n^2 \nabla^2 \phi_a + \cdots. \tag{15}$$

The propagators $\langle n\phi_a \rangle$ and $\langle nn \rangle$ can be obtained from the first line:

$$\langle n\phi_a \rangle(p) = \frac{1}{\omega + iDq^2}, \qquad \langle nn \rangle(p) = \frac{2T\sigma q^2}{\omega^2 + D^2 q^4}, \tag{16}$$

where $p = \{\omega, q\}$ denotes frequency and momentum collectively. The terms in the second line of (15) give cubic vertices: these lead to a density three point function (see Fig. 1):

$$\begin{aligned} G_{rrr}(p_2, p_1) = \; &T\sigma'(q_1 \cdot q_2)\langle n\phi_a \rangle(p_2)\langle n\phi_a \rangle(p_1)\langle nn \rangle(-p_1 - p_2) \\ &- iD'(q_1 + q_2)^2 \langle nn \rangle(p_2)\langle nn \rangle(p_1)\langle n\phi_a \rangle(-p_1 - p_2) \\ &+ 2 \text{ perm.}, \end{aligned} \tag{17}$$

where the permutations are obtained by swapping $p_1 \to p_2$, $p_2 \to -p_1 - p_2$ once, and twice. Higher-point functions can be similarly computed; the four-point function is obtained in (A.15).

## 3.1 Properties of the three-point function

There are a number of properties that the higher-point functions should satisfy, which serve as useful consistency checks of our calculation. First, notice that (17) vanishes when either $q_1$ or $q_2 \to 0$, as well as when $q_1 \to -q_2$; this must happen because the total charge $\int d^d x \, n(x, t)$ is conserved and has trivial dynamics. Second, other time orderings of the three-point function $G_{aar}$ and $G_{arr}$ can also be computed from the EFT (14): these should be related among each other and with $G_{rrr}$ through nonlinear KMS relations [16]. We check in App. A that these relations are satisfied. Third, in the limit where one of the densities is taken to be static, the three-point function reduces to the derivative of a two-point function with respect to chemical potential:

$$\lim_{q' \to 0} \lim_{\omega' \to 0} G_{aar}(p', p) = -i\frac{d}{d\mu}G_{ar}(p), \tag{18a}$$

$$\lim_{q' \to 0} \lim_{\omega' \to 0} G_{arr}(p', p) = -i\frac{d}{d\mu}G_{rr}(p), \tag{18b}$$

where $G_{ar}(p) = -i\sigma q^2 \langle n\phi_a \rangle(-p)$ and $G_{rr}(p) = \langle nn \rangle(p)$. Since $\frac{d}{d\mu} = \chi \frac{d}{dn}$, and $G_{rr}$, $G_{ar}$ depend on the background density through $\sigma(n)$, $D(n)$, Eq. (18) further elucidates why the three-point function depends on $\sigma'$ and $D'$; we verify that it holds in App. A.

## 3.2 Fourier transform

The Fourier transform of (17) can be evaluated, for direct comparison with numerics or experiments that probe nonlinear response in space and time domain.[4] Following Eqs. (6) and (7), we remove the overall $1/t^d$ scaling and directly extract the universal scaling functions $g_{3,D'}$ and $g_{3,\sigma'}$. In this section, we focus on $d = 1$ spatial dimension for simplicity. The scaling function $g_{3,\sigma'}$ is most simple and is given by

$$g_{3,\sigma'}\left(\frac{t_1}{t_2}, y_1, y_{21}\right) = \frac{1}{8\pi}e^{-\frac{1}{4}(y_1^2 + y_{21}^2)}, \tag{19}$$

with $y_1 \equiv x_1/\sqrt{Dt_1}$ and $y_{21} \equiv (x_2 - x_1)/\sqrt{D(t_2 - t_1)}$. The other scaling function $g_{3,D'}$ is much richer and is shown in (A.17). When all three points lie on the same site $y_1 = y_{21} = 0$, it simplifies to

$$g_{3,D'}\left(\frac{t_1}{t_2}, 0, 0\right) = \frac{1}{8\pi}\left(1 + 2\sqrt{\frac{t_1}{t_2}} + 2\sqrt{1 - \frac{t_1}{t_2}}\right). \tag{20}$$

Recall that we have taken $t_2 > t_1 > 0$. In this kinematics, the entire three-point function is given by

$$\langle n(t_2, 0)n(t_1, 0)n(0, 0)\rangle = \frac{(\chi T)^2}{8\pi D\sqrt{t_1(t_2 - t_1)}}\left[\frac{\sigma'}{\sigma} + \frac{D'}{D}\left(1 + 2\sqrt{\frac{t_1}{t_2}} + 2\sqrt{1 - \frac{t_1}{t_2}}\right)\right]. \tag{21}$$

An interesting kinematic configuration that neatly distinguishes both scaling functions is to take two of the points at equal time, $t_1 = t_2$, while keeping $x_1 \neq x_2$. There one finds from (19) that $g_{3,\sigma'}$ vanishes exponentially fast as $t_2 \to t_1$ for $x_1 \neq x_2$ and only produces a contact term $\propto \delta(x_1 - x_2)$ in this limit. The entire correlator at separated points is then proportional to $D'$:

$$\langle n(t, x_2)n(t, x_1)n(0, 0)\rangle = \frac{(\chi T)^2}{8\pi Dt}\frac{D'}{D}e^{-\frac{1}{4}(y_1^2 + y_2^2)}\left[1 - y_1 e^{\frac{1}{4}y_2^2}\left(\text{erf}\left(\frac{y_2}{2}\right) + \frac{\sqrt{\pi}}{2}\text{sign}(y_1 - y_2)\right)\right]$$
$$+ (y_1 \leftrightarrow y_2), \tag{22}$$

where $\text{erf}(s) = \int_0^s du\, e^{-u^2}$. The fairly long-range correlations at time $t$ in (22) are an equilibrium analog of long-range, equal-time correlations that arise in out-of-equilibrium states [26].

## 3.3 OPE in the EFT

Any EFT defined by a path integral satisfies an operator product expansion (OPE) [35]: two operators approaching each other can be approximated by an expansion in *local* operators of the EFT. This provides a useful organizing principle for nonlinear response in the EFT. For densities, this gives

$$n(x, t)n(0, 0) \sim \frac{\mathbb{1}}{(Dt)^{d/2}}f_{nn}{}^{\mathbb{1}}\left(\frac{x^2}{Dt}\right) + \frac{n(0, 0)}{(Dt)^{d/2}}f_{nn}{}^n\left(\frac{x^2}{Dt}\right) + \cdots, \quad \text{as} \quad x^2 \sim Dt \to 0, \tag{23}$$

where $\cdots$ contains contributions from higher dimension operators such as $\nabla n$, $n^2$, etc. The EFT OPE should be understood as valid in the limit where $n(x, t)$ and $n(0, 0)$ are much closer than any other two EFT operators, but still with a separation greater than the EFT cutoff. We have used the tree level scaling dimension $n(x, t) \sim q^{d/2}$ to obtain the scaling of the first term, and furthermore used the fact that second term requires the dimensionful couplings $q_\Lambda^{d/2}$ to

---

[4]Note that the correlator in mixed $t, q$ representation $\langle n(t_2, q_2)n(t_1, q_1)n(0, -q_1 - q_2)\rangle$ is expected to receive large loop corrections at late time due to the 'diffuson cascade' [33]. We therefore consider correlators in space and time $t, x$ in this section.

obtain the scaling of the second term. The first scaling function describing the $n \times n \to \mathbb{1}$ fusion channel can be simply read off the two point function (1): $f_{nn}^{\mathbb{1}}\left(\frac{x^2}{Dt}\right) = \frac{\chi T}{(4\pi)^{d/2}} e^{-\frac{x^2}{4Dt}}$. However, the next OPE function, describing the $n \times n \to n$ fusion channel is more interesting. It constrains a limit of the three-point function:

$$
\begin{aligned}
\lim_{x,t \to 0} \langle n(x',t')n(x,t)n(0,0) \rangle &= \frac{1}{(Dt)^{d/2}} f_{nn}^{\ n}\left(\frac{x^2}{Dt}\right) \langle n(x',t')n(0,0) \rangle \\
&= \frac{1}{(D^2 t t')^{d/2}} f_{nn}^{\ n}\left(\frac{x^2}{Dt}\right) f_{nn}^{\ \mathbb{1}}\left(\frac{x'^2}{Dt'}\right).
\end{aligned}
\tag{24}
$$

The fact that the dependence on $x,t$ and $x',t'$ factorizes is an obvious consequence of taking the limit $x^2 \sim Dt \ll x'^2 \sim Dt'$ – the nontrivial content of (24) is however that the dependence on $x'^2/(Dt')$ is entirely fixed. Eq. (19) shows that the $\sigma'$ piece satisfies this property. We show in App. A that the $D'$ piece of the three-point function satisfies it as well.

## 4 Applications

### 4.1 Numerics

We expect the scaling behavior (3) to apply to any diffusive many-body system, with the precise form of the scaling function (17) applying more specifically to systems with a single conserved density. This includes quantum and classical spin chains, random or deterministic unitary circuits and cellular automata with a conserved density, and lattice gases. As a simple test of our predictions, we study nonlinear response numerically in the Katz-Lebowitz-Spohn (KLS) model [36], a classical lattice gas with conserved particle number. One appeal of this model is that the diffusivity $D(n)$ and conductivity $\sigma(n)$ are known analytically as functions of the background density $n$ [37, 38] – all parameters entering our prediction for the three-point function in Eq. (17) or (21) are therefore fixed.

Let us first verify the universal scaling behavior of correlators from (3): Fig. 2 shows a clear $1/t^{(N-1)/2}$ decay of the $N$-point functions, for $N = 2, 3$. Moreover, the prefactor of this polynomial behavior agrees with the theory prediction from (21) at late times. As a more refined test of our results, we extract numerically the dimensionless scaling function $g_3$ (6) and compare it to the prediction (19) and (20) (insets of Figs. 2 and 3). In Fig. 2, the predicted scaling function is fairly featureless and difficult to confirm. To enhance its features, Eq. (21) suggests to study a region of parameters where the $D'$ coefficient is large; this is done in Fig. 3, where a good agreement with the predicted scaling function $g_3$ is found. A larger $D'$ is also known to produce larger loop corrections to the correlators [3] – Fig. 3 shows that these corrections are indeed appreciable at intermediate times.

### 4.2 Generalizations

We have focused on nonlinear response in diffusive systems $\omega \sim -iq^z$ with $z = 2$, because this is the most common dissipative universality class observed in thermalizing systems. However, there are other possibilities: superdiffusion in the KPZ universality class $z = \frac{3}{2}$ occurs for sound modes in $d = 1$ [32], and subdiffusion (e.g., $z = 4$) can arise in constrained systems with dipole-type symmetries [39–41]. One can extend a scaling argument similar to Eq. (3) for these situations: Assuming that nonlinear static susceptibilities are finite

$$
\langle n_r n_a \cdots n_a \rangle \sim \frac{d^{N-1} n}{d\mu^{N-1}} \sim 1,
\tag{25}
$$

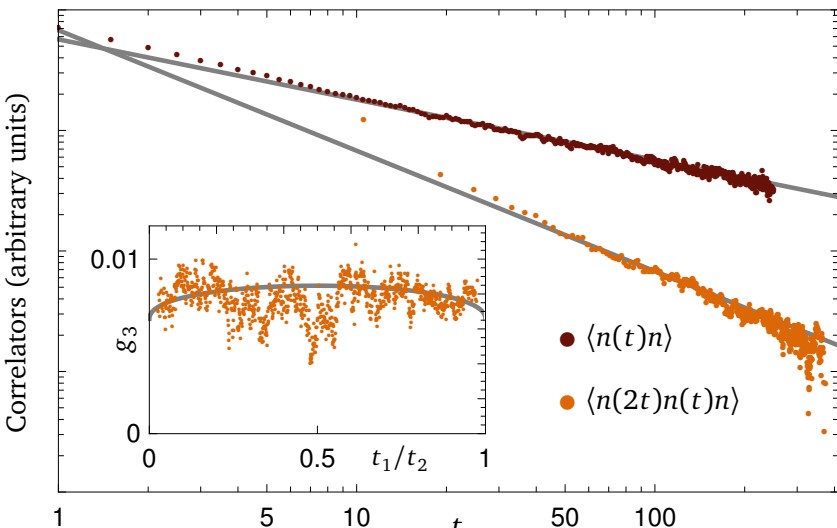

Figure 2: Log-log plot of the connected two- and three-point functions of density in the KLS model. The gray curves are the theory predictions from (1) and (21), with no fitting parameters. Inset: dimensionless scaling function (7) $g_3(\frac{t_2}{t_1}, x_{1,2} = 0)$ compared to theory (21). (Numerical parameters: $\delta = 0.9$, $L = 2^{19}$, $\langle n \rangle = 0.9$, averaged over 5 realizations).

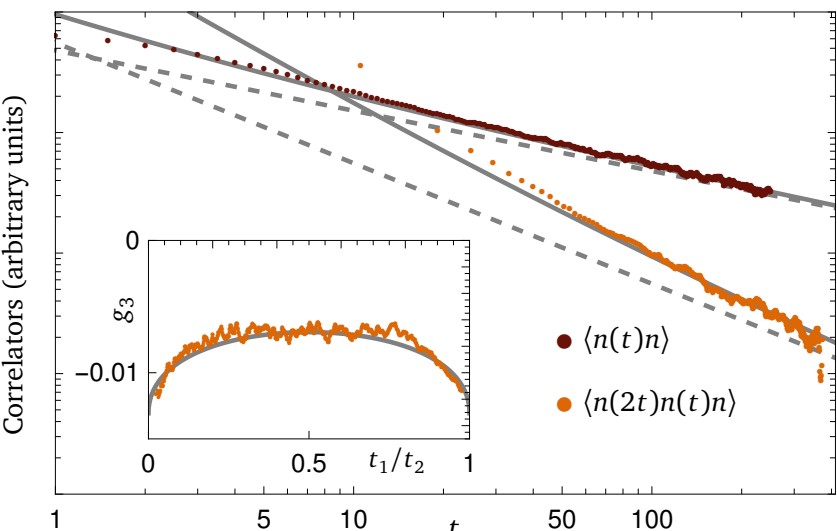

Figure 3: Same as Fig. 2, but with a filling $\langle n \rangle = 0.35$ chosen such that $D'$ is large, leading to a scaling function with clearer features (inset). A large $D'$ also leads to enhanced $1/\sqrt{t}$ corrections due to loops [3]. Dashed lines show the predictions from (1) and (21) without corrections, and solid lines include 1-loop corrections. For the two-point function, the exact prefactor of the 1-loop correction from [3] was used; for the three-point function, the corresponding computation is not available so the coefficient was fit. Fig. 4 shows the diagrams contributing to this correction.

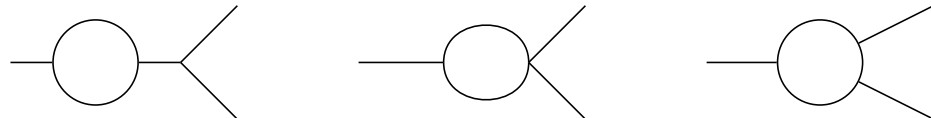

Figure 4: Loop corrections giving a relative $1/t^{d/2}$ correction to the three point function: $\langle nnn \rangle \sim \frac{1}{t^d}\left(1 + \frac{1}{t^{d/2}}\right)$. In $d \le 2$ this is the leading correction to the three-point function.

and that the entire $N$-point function is scaling according to $\omega \sim q^z$ in the hydrodynamic regime, one finds

$$\langle n((N-1)t)\cdots n(2t)n(t)n(0)\rangle \sim 1/t^{\frac{d}{z}(N-1)}. \tag{26}$$

The scaling assumption however does not apply to hydrodynamic regimes that include a scale to leading order in dissipation, including balistic modes $\omega(q) = c_s q - iDq^z$ with length scale $(D/c_s)^{1/(1-z)}$. The leading nonlinearities in these situations are non-dissipative (see, e.g., [23]), but the subleading diffusive regime can be captured by generalizing the scaling argument above: note that nonlinearities from $c_s(n)$ are $1/q^{z-1}$ enhanced compared to those from $D(n)$; using $N-2$ such vertices then leads to $\langle n\cdots nn \rangle \sim 1/t^{\frac{d(N-1)-(z-1)(N-2)}{z}}$ along the sound front.[5]

To go beyond this general scaling behavior and obtain the universal dimensionless scaling functions for nonlinear response, one would need to use the appropriate EFTs for the different hydrodynamic situations described above. It would be interesting to carry this out to allow for precision tests of thermalization in models exhibiting superdiffusion or subdiffusion.

# Acknowledgments

We thank Dmitry Abanin, Benjamin Doyon, Jacopo de Nardis, Paolo Glorioso, Mark Mezei, and Alexios Michailidis for inspiring discussions.

# A  Schwinger-Keldysh EFT for diffusion

**Nonlinear EFT**

The EFT of diffusion [4] consists in expressing the partition function (8) of a thermalizing system in terms of a local effective Lagrangian $\mathcal{L}$ containing 'Stückelberg' fields $\phi^I$:

$$Z[A_\mu^1, A_\mu^2] = \int D\phi_1 D\phi_2 \, e^{i \int dt \, d^d x \, \mathcal{L}[B_\mu^1, B_\mu^2]}, \tag{A.1}$$

with $B_\mu^I \equiv A_\mu^I + \partial_\mu \phi^I$ ($I = 1, 2$) and $\mu = 0, 1, \dots, d$ is a spacetime index. This is a minimal way of producing a gauge-invariant partition function; the number of degrees of freedom (two per symmetry) matches those of previous approaches to fluctuating hydrodynamics [31, 32] which contain one density and a noise field for each continuous symmetry. One then proceeds as usual in EFT by including in $\mathcal{L}$ all possible operators allowed by symmetries, in an expansion in derivatives and fields. There are several other conditions that the partition function must satisfy which leads to a number of constraints on $\mathcal{L}$ – we refer the reader to [28] for further details. Up to quadratic order in fields, one finds

$$\mathcal{L}^{(2)} = \sigma B_{ai}(iTB_{ai} - \dot{B}_{ri}) + \chi B_{a0}B_{r0} + \cdots, \tag{A.2}$$

---

[5]For the special case of the KPZ universality class $z = \frac{3}{2}$, which describes sound modes in $d = 1$, this becomes $1/t^{N/3}$, in agreement with [42].

where dot denotes a time derivative. The two undetermined Wilsonian coefficients $\sigma$ and $\chi$ will be found to correspond to the conductivity and susceptibility. The fact that the first two terms come with the same coefficient (up to a factor of the temperature $T$) follows from KMS relations. The ellipses denote higher derivative terms: these give $1/t$ corrections to any late-time observable and will be ignored here. Turning off the background fields, one has

$$\mathcal{L}^{(2)} = \sigma \nabla_i \phi_a (i T \nabla_i \phi_a - \nabla_i \dot{\phi}_r) + \chi \dot{\phi}_a \dot{\phi}_r + \cdots. \tag{A.3}$$

Balancing the three terms, one finds that the appropriate scaling is diffusive $\omega \sim q^2$, and $\phi_a \sim \dot{\phi}_r$. This leading scaling of the Gaussian action guides one in keeping only the most relevant nonlinear terms – at the cubic level these are:

$$\frac{1}{\chi} \mathcal{L}^{(3)} = \sigma' B_{r0} B_{ai} \left( i T B_{ai} - \dot{B}_{ri} \right) + \frac{1}{2} \chi' B_{a0} B_{r0}^2 + w F_{r,ij}^2 B_{a0} + \cdots. \tag{A.4}$$

Three new Wilsonian coefficients enter in the cubic action at leading order in derivatives: $\sigma'$, $\chi'$ and $w$. The last one involves the field strength $F_{r,ij}$ which is independent of the dynamical field $\phi_r$ – it will only produce contact term contributions to correlators involving the current, so we ignore it in the following. How suppressed are cubic terms compared to the quadratic action? Fluctuations in $\phi$ scale as

$$\int dt d^d x \, \mathcal{L}^{(2)} \sim 1 \quad \Rightarrow \quad \dot{\phi}_r \sim \phi_a \sim q^{d/2}. \tag{A.5}$$

The cubic action is therefore $\mathcal{L}^{(3)}/\mathcal{L}^{(2)} \sim q^{d/2}$ suppressed compared to the quadratic one. This is what allows for an expansion in fluctuations in the EFT: interactions are irrelevant in the RG sense. For example, they lead to $O(q^d) = O(\omega^{d/2})$ corrections to correlation functions [3] (because two cubic vertices are necessary). However, the cubic terms in (A.4) will give the leading contribution to three-point functions of densities or currents.

The density and current operators can be found by taking derivatives with respect to the background fields $j_r^\mu \equiv \delta \mathcal{L}/\delta A_{a\mu}$. For example, the density $n \equiv j_r^0$ is given by

$$n = \chi \left( B_{r0} + \frac{1}{2} \chi' B_{r0}^2 + \cdots \right). \tag{A.6}$$

Taking derivatives with respect to $\mu = A_{r0}$, one sees that $\chi = \frac{dn}{d\mu}$ indeed is the susceptibility, and $\chi' = \frac{1}{\chi} \frac{d\chi}{d\mu} = \frac{d\chi}{dn}$, justifying the notation. Similarly computing the current density $j_r^i$ shows that $\sigma$ is the dc conductivity, and $\sigma' = \frac{d\sigma}{dn}$.

To make contact with other approaches to fluctuating hydrodynamics, one can use this equation to trade the degree of freedom $\phi_r$ for $n$. The action up to cubic order then reads

$$\mathcal{L} = \left( \sigma + \sigma' n \right) B_{ai} (i T B_{ai} - E_{r,i}) + B_{a0} n - (D + D' n) B_{ai} \partial_i n + \cdots, \tag{A.7}$$

where we defined $D(n) \equiv \sigma(n)/\chi(n)$. This is the action used in the main text, see Eq. (15). One can similarly compute interactions involving more fields – to leading order in derivatives these simply take the form (14). The only Wilsonian coefficients involved are therefore the Taylor expansion coefficients of the diffusivity and conductivity (or susceptibility) around the density of interest (2). This agrees with other existing approaches to hydrodynamics [31,32], including macroscopic fluctuation theory [26] – we therefore expect that the results in this paper on the leading late time density three and four-point functions could be obtained from these other methods (to the best of our knowledge, they have not so far). However, we emphasize that the EFT approach of [4] allows to systematically compute corrections to this action (and therefore to observables such as the ones studied here), including terms that are missed in previous approaches. The leading such terms arise in the quartic action and give measurable, albeit small, corrections to any observable, see Ref. [5].

## Higher-point functions of densities

In this paper, we focus on higher-point functions of the density $n$. Correlators involving the longitudinal part of the current can be obtained using Ward identities. The transverse part of the current, instead, does not overlap with any long lived operator in theories with a single diffusive charge.[6] Its correlators are therefore pure contact terms in the EFT.

We define connected $N$-point functions of $r/a$ densities as

$$G_{r\cdots ra\cdots a}(1,\ldots,N) \equiv \frac{\delta^N \log Z}{\delta(iA_a^0(1))\cdots\delta(iA_a^0(N_r))\delta(iA_r^0(N_r+1))\cdots\delta(iA_r^0(N))}. \tag{A.8}$$

$N_r$ denotes the number of '$r$' operators, and $N-N_r$ the number of $a$ operators. Note that this definition slightly differs from that of [16] – the two are related as

$$G_{r\cdots ra\cdots a}^{[16]} = (-i)^{N-1} 2^{N_r-1} G_{r\cdots ra\cdots a}^{\text{here}}. \tag{A.9}$$

The three-point function $G_{rrr}$ is computed in the main text and expressed in terms of propagators in Eq. (17). Explicitly, it is given by

$$G_{rrr}(p_1,p_2) = (T\chi)^2 \left[\frac{2\sigma'}{\sigma} - \frac{4D'}{D}\right] \frac{k_1^2 k_2^2 k_3^2 (k_1^2+k_2^2+k_3^2)}{(\omega_1^2+k_1^4)(\omega_2^2+k_2^4)(\omega_3^2+k_3^4)} \tag{A.10}$$
$$-(T\chi)^2 \frac{4\sigma'}{\sigma} \frac{k_1^2(k_2\cdot k_3)\omega_2\omega_3 + k_2^2(k_3\cdot k_1)\omega_3\omega_1 + k_3^2(k_1\cdot k_2)\omega_1\omega_2}{(\omega_1^2+k_1^4)(\omega_2^2+k_2^4)(\omega_3^2+k_3^4)},$$

where $k_i \equiv \sqrt{D}q_i$ and we introduced $\omega_3 \equiv -\omega_1-\omega_2$ and $k_3 \equiv -k_1-k_2$ to make manifest the permutation symmetry. The two inequivalent operator orderings $G_{arr}$ and $G_{raa}$ can be similarly obtained – they are given by

$$G_{raa}(p_1,p_2) = \chi^2 \Bigg(\frac{D'}{D} \frac{k_1^2 k_2^2 k_3^2}{\left(k_1^2+i\omega_1\right)\left(k_2^2-i\omega_2\right)\left(k_3^2-i\omega_3\right)}$$
$$+ \frac{\sigma'}{\sigma} \frac{k_1 k_2 k_3 \left(k_3\left(k_1 k_2+i\omega_2\right)+ik_2\omega_3\right)}{\left(k_1^2+i\omega_1\right)\left(k_2^2-i\omega_2\right)\left(k_3^2-i\omega_3\right)}\Bigg), \tag{A.11a}$$

$$G_{arr}(p_1,p_2) = 2T\chi^2 \Bigg(\frac{iD'}{D} \frac{k_1^2 k_2^2 k_3^2 \left(k_2^2+k_3^2-i\omega_2-i\omega_3\right)}{\left(k_1^2-i\omega_1\right)\left(k_2^4+\omega_2^2\right)\left(k_3^4+\omega_3^2\right)}$$
$$+ \frac{i\sigma'}{\sigma} \frac{k_1 k_2 k_3 \left(k_2\omega_3^2 - 2k_1\omega_2\omega_3 + k_3\left(\omega_2^2 - k_2 k_1\left(k_2^2+k_3 k_2+k_3^2\right)\right)\right)}{\left(k_1^2-i\omega_1\right)\left(k_2^4+\omega_2^2\right)\left(k_3^4+\omega_3^2\right)}\Bigg). \tag{A.11b}$$

The momenta $p_1, p_2$ are carried by the first two arguments of the Green's functions. For example, $\frac{\delta^3 \log Z}{\delta(iA_r^0(p_1))\delta(iA_a^0(p_2))\delta(iA_a^0(p_3))} \equiv (2\pi)^{d+1}\delta^{d+1}(p_1+p_2+p_3)G_{arr}(p_1,p_2)$.

One can check that these satisfy nonlinear KMS relations [16], which to the order we are working in read

$$\text{Re}(G_{arr}+G_{rar}+G_{rra}) = 0, \tag{A.12a}$$

$$\text{Im}(G_{rrr}) = 0, \tag{A.12b}$$

$$\text{Re}(G_{rrr}) = -\frac{T}{\omega_1}\text{Re}(-G_{arr}+G_{rar}+G_{arr}). \tag{A.12c}$$

---

[6]This is not the case in the presence of two diffusive modes [43].

In a certain limit, the three-point functions above should reduce to derivatives of two-point functions with respect to chemical potential (18). From (A.11), one finds that this is indeed the case

$$
i \lim_{\omega_1 \to 0} \lim_{p_1 \to 0} G_{raa}(p_1, p_2) = \frac{i\chi^2 k_2^2}{(k_2^2 - i\omega_2)^2} \left( \frac{D'}{D} k_2^2 - \frac{\sigma'}{\sigma} (k_2^2 - i\omega_2) \right) = \frac{d}{d\mu} G_{ra}(p_2),
$$
$$
i \lim_{\omega_1 \to 0} \lim_{p_1 \to 0} G_{arr}(p_1, p_2) = \frac{-2T\chi^2 k_2^2}{(k_2^4 + \omega_2^2)^2} \left( \frac{2D'}{D} k_2^4 - \frac{\sigma'}{\sigma} (k_2^4 + \omega_2^2) \right) = \frac{d}{d\mu} G_{rr}(p_2).
$$

(A.13)

In systems with charge conjugation symmetry, such as particle-hole symmetric spin chains at half filling, cubic nonlinearities $D'$, $\sigma'$ vanish. In these situations, the leading non-Gaussianities are quartic:

$$
\mathcal{L}^{(4)} = \frac{1}{2} \sigma'' n^2 B_{ai} (iTB_{ai} - E_{r,i}) - \frac{1}{2} D'' n^2 B_{ai} \partial_i n + \cdots,
$$

(A.14)

leading to the following four-point function

$$
G_{rrrr}(p_3, p_2, p_1) = 2T\sigma''(q_1 \cdot q_2) \langle nn \rangle(p_3) \langle n\phi_a \rangle(p_2) \langle n\phi_a \rangle(p_1) \langle nn \rangle(p_4) + 5 \text{ perm.}
$$
$$
- iD'' q_4^2 \langle nn \rangle(p_3) \langle nn \rangle(p_2) \langle nn \rangle(p_1) \langle n\phi_a \rangle(p_4) + 3 \text{ perm.},
$$

(A.15)

where we introduced $p_4 = -p_1 - p_2 - p_3$ to simplify notation. In the absence of charge conjugation symmetry, the four-point function also receives contributions proportional to $D'^2$, $D'\sigma'$ and $\sigma'^2$ from the vertices in Fig. 1.

## Fourier transform and OPE

For comparison with experiments or numerics working in the time and space domain, we Fourier transform the density three-point function $\langle nnn \rangle(p_1, p_2) = G_{rrr}(p_1, p_2)$ found in (17) or (A.10). For simplicity, we focus on $d = 1$ spatial dimensions. Specifically, we compute

$$
\langle n(t_2, x_2) n(t_1, x_1) n(0, 0) \rangle = \int \frac{d\omega_1}{2\pi} \frac{d\omega_2}{2\pi} \frac{dq_1}{2\pi} \frac{dq_2}{2\pi} e^{-i(\omega_1 t_1 + \omega_2 t_2 - q_1 x_1 - q_2 x_2)} G_{rrr}(p_1, p_2), \quad (A.16)
$$

assuming $t_2 \geq t_1 \geq 0$. The first two frequency integrals can be straightforwardly computed by residues; the final two integrals require a little more work but can be evaluated as well. One finds that the three-point function takes the form (6), (7) as expected, with $g_{3,\sigma'}$ shown in (19), and

$$
8\pi g_{3,D'} = e^{-\frac{1}{4}(y_1^2 + y_2^2)} \sqrt{1-A} \left[ 2 - \sqrt{A} e^{\frac{1}{4}y_1^2} y_2 \operatorname{erf}(\tfrac{y_1}{2}) - \frac{1}{\sqrt{A}} e^{\frac{1}{4}y_2^2} y_1 \operatorname{erf}(\tfrac{y_2}{2}) \right]
$$
$$
+ e^{-\frac{1}{4}(y_2^2 + y_{21}^2)} \sqrt{A} \left[ 2 - \frac{1}{\sqrt{1-A}} e^{\frac{1}{4}y_2^2} y_{21} \operatorname{erf}(\tfrac{y_2}{2}) - \sqrt{1-A} e^{\frac{1}{4}y_{21}^2} y_1 \operatorname{erf}(\tfrac{y_{21}}{2}) \right] \quad (A.17)
$$
$$
+ e^{-\frac{1}{4}(y_1^2 + y_{21}^2)} \left[ 1 + \frac{\sqrt{1-A}}{\sqrt{A}} e^{\frac{1}{4}y_{21}^2} y_1 \operatorname{erf}(\tfrac{y_{21}}{2}) + \frac{\sqrt{A}}{\sqrt{1-A}} e^{\frac{1}{4}y_1^2} y_{21} \operatorname{erf}(\tfrac{y_1}{2}) \right],
$$

with $y_i \equiv x_i / \sqrt{Dt_i}$, $y_{21} \equiv (x_2 - x_1)/\sqrt{D(t_2 - t_1)}$ and $A \equiv t_1/t_2$, and with $\operatorname{erf}(s) \equiv \int_0^s du\, e^{-u^2}$. The scaling function only depends on three dimensionless ratios of coordinates which can be taken to be $t_1/t_2$, $y_1$ and $y_2$, but $y_{21}$ was introduced to simplify the final expression. Various limits of this expression lead to Eqs. (21) and (22) quoted in the main text.

One can show that the three-point function found above satisfies the factorization property (24), and compute the OPE function $f_{nn}{}^n$. First, we expect two independent OPE scaling functions related to the two leading nonlinearities of the EFT

$$
f_{nn}{}^n \left( \frac{x^2}{Dt} \right) = \chi T \frac{D'}{D} f_{nn}{}^n|_{D'} \left( \frac{x^2}{Dt} \right) + \chi T \frac{\sigma'}{\sigma} f_{nn}{}^n|_{\sigma'} \left( \frac{x^2}{Dt} \right).
$$

(A.18)

The two OPE functions $f_{nn}{}^n|_{D'}$ and $f_{nn}{}^n|_{\sigma'}$ are simplified limits of the scaling functions defined in Eq. (7). One finds

$$f_{nn}{}^n|_{\sigma'}\left(y^2\right) = \frac{e^{-\frac{1}{4}y^2}}{\sqrt{4\pi}}, \qquad f_{nn}{}^n|_{D'}\left(y^2\right) = \frac{e^{-\frac{1}{4}y^2}}{\sqrt{4\pi}}\frac{y^2-6}{4}. \tag{A.19}$$

Fourier transforming back, one also expects the OPE to control the large $\omega, q$ limit of correlators, so that the 3pt function must have the limit

$$\lim_{\omega, q \to \infty} G_{rrr}(\omega', q', \omega, q) = \frac{1}{\omega\omega'}\tilde{f}_{nn}{}^n\left(\frac{\omega}{Dq^2}\right)\tilde{f}_{nn}{}^{\mathbb{1}}\left(\frac{\omega'}{Dq'^2}\right), \tag{A.20}$$

where $\tilde{f}_{nn}{}^{\mathbb{1}}\left(\frac{\omega'}{Dq'^2}\right)$ is proportional to the density two-point function. We have checked that this is the case.

# B Details on the numerics

To test the predictions of the EFT, we consider a classical chain of bits, with charge at a site $i$ ($i = 1, \dots, L$) taking two possible values $n_i \in \{0, 1\}$:

$$\cdots 0\,1\,0\,1\,1\,0\,0\,1\,0\,0\,1\cdots. \tag{B.1}$$

The dynamics described below will be local, satisfy detailed balance, and will conserve the total charge $\sum_i n_i$ – if it thermalizes it should therefore be described by the EFT of diffusion.

The dynamics we consider on this system is the Katz-Lebowitz-Spohn model [36]. It consists in a Monte-Carlo type time evolution where at every step, a site $i$ is randomly chosen, and the following '4-site gate' is applied to the charges $n_{i-1}\,n_i\,n_{i+1}\,n_{i+2}$:

$$0\,1\,0\,0 \quad \xrightarrow{r(1+\delta)} \quad 0\,0\,1\,0, \tag{B.2a}$$

$$1\,1\,0\,1 \quad \xrightarrow{r(1-\delta)} \quad 1\,0\,1\,1, \tag{B.2b}$$

$$1\,1\,0\,0 \quad \xrightarrow{r(1+\epsilon)} \quad 1\,0\,1\,0, \tag{B.2c}$$

$$0\,1\,0\,1 \quad \xrightarrow{r(1-\epsilon)} \quad 0\,0\,1\,1, \tag{B.2d}$$

where the number above the arrow denotes the probability $P$ of the transition happening. The particle at $n_i$ therefore hops to the right with probability depending on the configuration of is neighbors. Processes related to those shown above by reflection occur with equal probability. The model depends on two parameters $\delta$ and $\epsilon$. We will set $\epsilon = 0$ below: in this case the rates above clearly satisfy infinite temperature detailed balance, $e^{-\beta H} = 1$ (the model with $\epsilon \neq 0$ can also be shown to satisfy detailed balance [36, 44]). Time will be measured in units of the rate $r$, which will therefore not appear below. We take a single time step $\Delta t = 1/r$ to correspond to $L$ applications of the gate above.

An appealing feature of this model for the purposes of testing the EFT predictions is that $D(n)$ and $\sigma(n)$ are known analytically as a function of the density measured in units of the lattice constant $n = \frac{1}{L}\sum_i n_i \in [0, 1]$, and the parameters of the model $\delta, \epsilon$ [37, 38]. The coefficients of the nonlinear EFT (15) to leading order in gradients are therefore entirely fixed. For our purposes, it will be sufficient to consider the model with $\epsilon = 0$, in which case these are given by (note that $T\chi$ is finite in the limit $T \to \infty$)

$$T\chi(n) = n(1-n), \qquad D(n) = 1 + \delta(1-2n). \tag{B.3}$$

From these expressions, one can obtain the cubic interactions $D'$ and $\sigma'$ appearing in the EFT. Figs. 2 and 3 show the numerical results for correlation functions, and the corresponding predictions from the EFT, using the values $D'$ and $\sigma'$ obtained above.

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
