# Peer review of "Nonlinear Response in Diffusive Systems"

_SciPost Physics, doi:SciPost Phys. 16, 047 (2024)_

## Round 1 · Referee Report · Anonymous (Referee 1) · 2023-12-30

Report

The authors study a particular N-point function of a conserved charge density in diffusive systems. A general form for a nonlinear response function at late times is proposed and several consistency checks are performed. In particular, the result for the scaling function determining the density 3-point function is analyzed in some detail. Fluctuation fields, which are systematically described using an EFT approach, contribute to these correlators and to subleading corrections. The scaling laws are confirmed numerically in a simple spin chain model.

Unfortunately, the 1-loop correction to the 3-point function is not actually computed, but only its scaling behavior is confirmed on general grounds. This leaves the analysis a bit qualitative. Nevertheless, this is an interesting step forward, as the tree-level results already seem to be new.

I think this is a paper that will be of interest to the community and should be published in SciPost. The presentation is very good and I only suggest minor improvements (mostly optional).

Requested changes

1- Eq. (24) could use a more careful explanation. Is a limit of small x'^2 and t' taken in order to apply the OPE (23)? If yes then how to understand the inequality in the text after (24)?

2- In section 3 the effective action approach of Crossley-Glorioso-Liu [4] is central. I feel that the references are a bit sparse here. As far as I understand, parallel works on the topic such as 1803.11155 (or 1804.04654) are largely isomorphic and lead to the same EFT of fluctuating hydrodynamics, certainly as far as the specific terms analyzed in this manuscript are concerned. Correct? If so, it might be worth referencing this properly.

3- Typos in the denominator of eq. (21) and in last sentence of section 4.1.

4- The detailed discussion of numerics and scaling functions is only done for d=1 spatial dimensions and in fact for $x_i=0$. This seems quite special. Is it possible to check spatial dependence (or higher dimensions) more nontrivially?

  • validity: good
  • significance: high
  • originality: good
  • clarity: top
  • formatting: perfect
  • grammar: perfect

Author:  Luca Delacrétaz  on 2024-01-26  [id 4284]

(in reply to Report 1 on 2023-12-30)

Response to report 1

We thank the referee for their careful reading of the manuscript and comments.

  1. Eq. (24) does not need x',t' -> 0 -- instead it is n(x,t) which fuses with the density at the origin n(0,0). This only requires x,t->0. We have added the coordinate (0,0) on the third density to avoid confusion.

  2. While we focus on diffusion in this work, we have added references to other EFT approaches to fluid dynamics for completeness.

  3. The typos have been fixed.

  4. Recent numerical work (https://arxiv.org/abs/2310.10564, Sec. V) has found quantitative agreement with the EFT prediction for the the spatial dependence of the three point function $x_i\neq 0$, confirming our Eq. (22). Numerical tests in higher dimensions remain to be done -- these should be possible for classical systems, albeit costly.

Author:  Luca Delacrétaz  on 2024-01-24  [id 4278]

(in reply to Report 1 on 2023-12-30)

We thank the referee for their careful reading of the manuscript and comments.

  1. Eq. (24) does not need x',t' -> 0 - instead it is n(x,t) which fuses with the density at the origin n(0,0). This only requires x,t->0. We have added the coordinate (0,0) on the third density to avoid confusion.

  2. While we focus on diffusion in this work, we have added references to other EFT approaches to fluid dynamics for completeness.

  3. The typos have been fixed.

  4. Recent numerical work (https://arxiv.org/abs/2310.10564, Sec. V) has found quantitative agreement with the EFT prediction for the the spatial dependence of the three point function $x_i\neq 0$, confirming our Eq. (22). Numerical tests in higher dimensions remain to be done - these should be possible for classical systems, albeit costly.

---

## Round 1 · Referee Report · Anonymous (Referee 2) · 2024-1-3

Strengths

See below.

Weaknesses

See below.

Report

This is a very interesting and well-written paper describing a basic observable in fluctuating hydrodynamics, the three-point function in nonlinear response. Interestingly, it appears that this quantity had never been thoroughly investigated before. The authors present a careful discussion of scaling relations and then use the recent effective field theory of fluctuating hydrodynamics to explicitly construct the three-point function, including a universal scaling function. They then compare this to numerical simulations of a simple classical hydrodynamic system, displaying an impressive agreement (modulo a small caveat below). This is a great application of the EFT formalism. I think this is a great paper and it certainly deserves to be published. I have some small suggestions:

  1. The description of the classical lattice gas in Appendix B describes a rule for time evolution. Below Eq B.1 it is said that this rule is local and conserves a charge, and therefore must be described by the EFT for diffusion. This isn't obvious to me; it seems that it is implicit in the construction of the EFT that the system relaxes to a thermal equilibrium, which (I believe) would generically happen only if the dynamics respects detailed balance for an underlying Hamiltonian. This isn't obvious from the evolution rule in B.1 alone -- however after taking a look at Ref [34] it seems to me that actually this dynamics in B.1 does respect detailed balance, but it would be nice for the authors to explain this somewhat further in the Appendix and perhaps mention the Hamiltonian (In particular, if I have misunderstood and this is not the case then the authors should explain why the EFT correctly describes the dynamics).

  2. This is not required: it doesn't seem like the diagrams in Figure 4 are that complicated, and computing them would greatly strengthen the agreement in Figure 3 (by removing the fitting parameter). Is this an option? I presume it isn't straightforward; if this is the case there is no need to do it this work.

Requested changes

See above.

  • validity: top
  • significance: high
  • originality: top
  • clarity: top
  • formatting: perfect
  • grammar: perfect

Author:  Luca Delacrétaz  on 2024-01-24  [id 4277]

(in reply to Report 2 on 2024-01-03)

We thank the referee for their careful reading of the manuscript and comments.

  1. The referee is correct - we expect the EFT to describe the emergent late time dynamics of many body systems if they are local, non-integrable, have a conservation law and satisfy detailed balance. While this last condition is automatic for systems undergoing Hamiltonian evolution, it must be imposed by hand for stochastic systems. In our case (epsilon=0 in the KLS model), the dynamics satisfy detailed balance for infinite temperature $\beta H = 0$. We have added several clarifying comments in Appendix B.

  2. We agree that evaluating the three diagrams in Fig. 4 (along with their 5 permutations) - and then Fourier transforming the result - could strengthen the quantitative agreement with numerics shown in Fig. 3, by removing the fitting parameter used in that figure. These corrections are also expected to feature interesting non-analyticities in momentum space as a function of the four variables w,k,w',k'. We however leave this lengthy calculation for the future. In the meantime, lack of analytic predictions for these power-law corrections can be remedied by more clever numerics: Fig. 2 shows comparisons without any fitting parameters in a situation where these loop corrections are suppressed (because D' is small).

---

## Round 2 · Author Response

We thank both referees for their detailed comments, which we address in our replies. Following their suggestions, we have made changes to the manuscript listed below.

---

## Round 2 · List of Changes

• Added a several clarifications concerning detailed balance in Appendix B.

  • Typos fixed, in particular in Eqs. (20), (21).

  • Added coordinate (0,0) in Eq. (24) to avoid confusion.

  • References added

---

## Editorial Decision

published